# Indigenous Voices Against Suicide: A Meta-Synthesis Advancing Prevention Strategies

**DOI:** 10.3390/ijerph20227064

**Published:** 2023-11-15

**Authors:** Meenakshi Richardson, Sara F. Waters

**Affiliations:** 1Prevention Science Program, Washington State University Vancouver, Vancouver, WA 98686, USA; 2Department of Human Development, Washington State University Vancouver, Vancouver, WA 98686, USA; sara.f.waters@wsu.edu

**Keywords:** suicidality, Indigenous, meta-synthesis, culturally grounded, prevention science

## Abstract

Rates of suicidality amongst Indigenous Peoples are linked to historical and ongoing settler-colonialism including land seizures, spiritual oppression, cultural disconnection, forced enculturation, and societal alienation. Consistent with decolonial practices, Indigenous voices and perspectives must be centered in the development and evaluation of suicide prevention programs for Indigenous Peoples in the United States to ensure efficacy. The current study is a meta-synthesis of qualitative research on suicide prevention among Indigenous populations in the United States. Findings reveal little evidence for the centering of participant voices within existing suicide prevention programs. Applied thematic analysis of synthesis memos developed for each article in the final sample surfaced four primary themes: (1) support preferences; (2) challenges to suicide prevention; (3) integration of culture as prevention; and (4) grounding relationships in prevention. The need for culturally centered programming and the inadequacy of ‘pan-Indian’ approaches are highlighted. Sub-themes with respect to resiliency, kinship connection, and safe spaces to share cultural knowledge also emerge. Implications of this work to further the decolonization of suicide prevention and aid in the promotion of culturally grounded prevention science strategies are discussed.

## 1. Introduction

Rates of suicidality amongst American Indian and Alaska Native (AI/AN) and Native Hawaiian (NH) populations are staggering. Suicide is the eighth leading cause of death among AI/ANs across all ages, and the leading cause of death for NHs and Pacific Islanders ages 15–24 [1,2]. Ongoing societal alienation, land seizures, spiritual oppression, cultural disconnection, and forced enculturation inflicted upon AI/AN and NH populations contribute to elevated risk for suicide [3]. The majority of health research is conducted from a Eurocentric worldview and within positivist structures [4]. The evidence-based literature that applies decolonizing methodologies and practice is minimal, although interest in this area is growing [5]. Therefore, examining available research along with community collaboration that is published and available in the current literature can inform suicide prevention efforts. Preventative programs that aim to mitigate suicide risk within AI/AN and NH communities must consider healing historical trauma and contribute to integrative solutions that reflect culture, traditions, values, and diversity [6,7]. To do this, the voices and perspectives of community members must be centered in program development and evaluation. In the current study, a qualitative meta-analysis, referred to as a meta-synthesis, of evidence-based suicide prevention programs was conducted to identify the extent to which community and participant voices are included to inform suicide prevention strategies.

### 1.1. The Prevalence and Incidence of Suicide

Compared to the United States (US) general population, AI/AN peoples are disproportionately impacted by suicide. Compared to the US overall suicide rate of 12.08 per 100,000 people, the AI/AN population experiences suicide at a rate of 16.93 per 100,000 people [1]. Rates for suicide continue to surge. In 2018, Leavitt and collaborators in 2018 conducted a study that involved 18 states from the National Violent Death Reporting System. The data revealed that amongst AI/ANs, the rate of suicide was 21.5 per 100,000, a rate 3.5 times higher than all other populations [8,9]. The Hawaii State Health Department vital statistics report that the overall suicide death rate in the state is 12.9 per 100,000, while the disaggregated rate for NH and other Pacific Islander groups is 33.9 per 100,000, the highest of any other racial/ethnic group in the state [10]. Suicide rates amongst NH and other Pacific Islander groups have been reported among males to be 19.8 per 100,000 [11]. It would be negligent to not address that these suicide rates have been directly linked to the historical and ongoing impacts of settler-colonialism. 

### 1.2. Impacts of Colonization on Suicide Risk

Causal factors for suicide have been linked to historical trauma as a result of settler-colonialism, which severely influences the health of AI/AN and NH populations [5,12,13]. Colonization involved the mass genocide of Indigenous people as well as their forced removal and geographical relocation, forced enrollment of their children into boarding schools, and the prohibition of religious practices and cultural expression. In other words, colonization has resulted in loss of life, connection to land, place, identity, and traditional knowledge (i.e., values, traditions, language, etc.) as well as disruption of family systems. The result of these experiences is historical trauma, the “cumulative emotional and psychological wounding over the lifespan and across generations, emanating from massive group trauma experiences” [14] (p. 7). The historical trauma response involves severely adverse impacts on physical, behavioral, emotional, social, and spiritual health. The impact of historical trauma continues through intergenerational transmission [15]. Historical and current trauma are significant predictors of the health disparities experienced by AI/AN and NH people, especially concerning suicidality [16,17]. 

Settler-colonialism is not a thing of the past but permeates every aspect of our contemporary society. Forced acculturation of Indigenous communities continues in the present with AI/AN and NH people consistently facing acts of discrimination and political silencing as well as an overall lack of belonging. AI/AN and NH people face consistent and ongoing acts of prejudice and discrimination, such as racist sports mascots, as well as lack of belonging, representation access, and support in social systems such as schools, healthcare, and child welfare [18,19,20,21]. The extent to which Indigenous societies continue to assimilate to settler-colonial (white dominant) culture determines the extent to which acculturative, socioeconomic, and psychological stress is experienced both individually and intergenerationally [13,22]. Amongst the Yup’ik peoples in Alaska, greater connection to one’s cultural identity has been linked to the ability to healthily cope with stressors and regulate emotions [23]. Among the self-identified Yu’pik, those who reported having adapted to the mainstream US culture also reported decreased happiness, increased psychosocial stress, and increased substance use to cope with the loss of cultural identity and challenges of navigating within and between two very different worlds. In another study involving qualitative analysis of suicide notes left by AI/AN individuals, themes of alienation, grief, and social and emotional disconnection from one’s cultural identity were prevalent [24]. Hereafter, AI/AN and NH peoples will collectively be referred to as Indigenous or Indigenous Peoples.

### 1.3. The Current Study

There is a call in the field for approaches that emphasize protective factors, such as enriched cultural identity and social connectedness, to strengthen suicide prevention efforts among Indigenous Peoples [6]. There is a gap in the literature that identifies the extent to which prevention programs addressing suicidality are based on Indigenous Ways of Knowing and center Indigenous people’s stories and lived experiences, instead of western approaches to suicide prevention [4,25]. Tuck and Yang emphasize the need for decolonization to centralize the reciprocal relationships between Indigenous peoples, their land, and wellbeing. Employing qualitative and data storytelling uproots settler-colonial perspectives to encompass holistic methodologies that reinforce social justice and preventative pedagogy consistent with cultural praxis [26]. The current study offers insight into qualitative suicide interventions and prevention programs for Indigenous Peoples through a qualitative meta-analysis, referred to as a meta-synthesis, of evidence-based suicide prevention programs for Indigenous Peoples to answer the following research questions: (Q1) To what extent are participant perspectives integrated to inform program development? and (Q2) What patterns and contradictions emerge among the qualitative analyses of stories and narratives conducted by the author? 

## 2. Materials and Methods

### 2.1. Theoretical Framework

Practices grounded in traditional knowledge and Indigenous knowledge systems as a theoretical framework, also referred to as Indigenous Ways of Knowing. Indigenous Ways of Knowing values respect, relationship, and reciprocity [4]. Indigenous Ways of Knowing fuel opportunities for participation, reflection, renewal, and healing by prioritizing community-based action research through the sharing of stories and embracing the value of both the individual and collective voice of Indigenous Peoples [27,28] is a process that disrupts the effects and perpetuation of colonization [29]. Decoloniality, often referred to as decolonization, has been used within the research community as an attempt to culturally ground strategies and approaches outside of positivist and settler-colonial oppression. It is imperative to note that prevention programs addressing suicide founded upon culturally grounded approaches and decolonizing methodologies have been proven successful, and acknowledging these works has the potential to encourage others to move in the same direction toward healing.

### 2.2. Author Positionality

The first author is a citizen of the Haliwa-Saponi Tribe residing in the Pacific Northwest. She has walked alongside tribal and urban Indian communities for nearly 9 years to provide health and human services, culturally informed research, and community-based programming. As both an Indigenous and Asian Indian woman, she was raised with rich languages, traditional knowledge, and cultural education. Her experiences with suicidal ideation during her youth and the impact of suicide and loss of kinship have led her toward serving Indigenous Peoples by engaging in transformative research and programming to highlight culture and resiliency. The second author is a white woman, mother, daughter, and sister living in the Pacific Northwest. Her western education has focused on developmental science, and child–caregiver relationships in particular, as well as prevention science and program development. Her ongoing learning occurs through longstanding relationships working alongside Indigenous students and partnering with a Northwest Tribe to support culturally grounded research to heal intergenerational trauma.

### 2.3. Qualitative Meta-Analyses

Qualitative analyses yield a multivocal story reflective of unique individual and shared experiences and is rarely employed with the intent that its findings be broadly generalized [30]. It may be used to highlight mechanisms of transferability to specific populations and those strictly reviewing the analyses in hopes of expanding on current knowledge, engage community voices, and foster sustainable relationships toward reconciliation and healing in research. Levitt explains that the purpose of a meta-analyses of the qualitative literature, or interchangeably referred to as meta-syntheses, is “to aggregate findings and identify patterns across primary studies” [31] (p. 367). A meta-synthesis involves a dynamic secondary analysis of primary qualitative data to provide a comprehensive and interwoven story that addresses a specific outcome across disparate studies [32]. Furthermore, a meta-synthesis can benefit from triangulation and representativeness to the sample (i.e., primary studies sources), and in this case, to further review a combination of methods and practices assessing suicidality specific to Indigenous samples to further validate culturally informed programming, research collaborations, and enrich decolonization efforts. 

### 2.4. Eligibility Criteria

All studies assessing suicide interventions and programs amongst Indigenous Peoples were eligible for review. The inclusion criteria were: (i) published in a scholarly peer-reviewed journal, (ii) written in English, (iii) reported on a suicide intervention or program, (iv) involves Indigenous populations in the US exclusively. There were no exclusion criteria based on publication date. 

### 2.5. Sources and Search Protocol 

Article searches were conducted in the PsychINFO, PubMed/MEDLINE, and SAGE Journals. The search protocol involved search strings, or word combinations categorically separated into population descriptors and content terms, combined with Boolean operators (e.g., American Indian ‘AND’ suicide intervention) [33].

### 2.6. Study Selection Process

Per Preferred Reporting Items for Systematic Reviews and Meta-analysis guidelines (PRISMA) [34], articles identified by all combinations of search terms from each database were exported into the Zotero reference software system and then unduplicated. The start set of publications to review was determined through a screening and filtration process. Filter 1 screened titles for relevance, which was defined as suicidality interventions or programs implemented exclusively with Indigenous Peoples. Filter 2 involved the review of abstracts of all preliminarily screened titles of programs or intervention for the inclusion of qualitative methodologies, whether mixed-methods or exclusively qualitative in nature. The articles remaining upon title and abstract screenings underwent Filter 3, a full article review. Within this filter the inclusion of action research was determined. Articles that did not meet all relevant characteristics after full article screening were excluded. Filters 1–3 were conducted by two reviewers (the first author and a research assistant) and cataloged to include titles, authors, abstracts, and full text publications, to track inclusion and exclusion decisions [33]. Conferencing meetings occurred between coders to discuss rationale and reach consensus. The second author was designated as a conflict resolver, if the two coders came to a disagreement. However, conflict resolution was not needed at any point throughout the analysis due to thorough conferencing. The remaining articles after screening and filtering were included as the main sample for further analysis. The screening and filtering are visually represented, and final sample size outlined within a PRISMA flowchart (Figure 1). 

### 2.7. Data Analysis 

To identify whether community perspectives were integrated into the intervention and programs included in this analysis per Q1, each article was assigned one of the following category codes: (1) interviews were conducted with participants; (2) focus groups, learning labs or equivalent were conducted with participants; (3) a community advisory board, group, or committee was included; (4) inclusion of tribal leadership (if tribal-specific); (5) authors explicitly note how participant perspectives were included although none of the prior categories were found in the article; or (6) no indication of the engagement of participant perspectives informing program development was outlined. To address patterns among qualitative analyses and differences in patterns for distinct groups per Q2, a synthesis memo was developed for each article in the sample. A synthesis memo is a formal summarization that can be used to distill information relevant to answering research questions [33]. Synthesis memos outline with headings for data capture, the date of the synthesis creation, full citation, main idea(s) or main theme(s) from the article, a summary memo which include a written summary, and an author summary which comprises direct quotes that explain the intervention, program purpose, and themes as reported in the article(s). There is also an evidence section, where quotes (either direct from participants or from the written components of an article), findings, and practice implications are outlined. These thorough synthesis memos served as the synthesis data that were coded for further analysis. 

### 2.8. Coding of Memos

Once the assortment of evidence was completed and established into synthesis memos for each article in the sample, a delineation of meaning was inspected to parse out categorical data [35] to formulate an overarching set of priori codes. Coding was employed through a hybrid approach that uses both inductive and deductive coding strategies [36]. Hence, codes were not developed a priori [37]. A codebook was developed to include both primary codes, which are umbrella codes that encompass main idea(s) and secondary codes, which are sub-codes that fall under primary codes. The codebook acted as a living document to keep track of the iterative coding process, outlining primary and secondary codes, definitions, or examples for each code [38]. 

### 2.9. Applied Thematic Analysis 

Further categorization and comparison emerging from the analysis was subjected to applied thematic analysis (ATA). ATA is an approach that is a way to identify and examine themes surfaced from textual data and can be employed for both primary and secondary data to comprehensively present the stories and experiences through the voice of participants [39]. ATA was employed to identify patterns or contradictions, as noted in Q2 to identify any connections between the current literature and the synthesis of the sample studies in an abductive process [40,41]. 

## 3. Results

Following Filters 1, 2, and 3, the final sample of articles that underwent analyses was *N* = 15 (see Figure 1 for full details). Publication year of sample articles ranged from 2006–2022, a majority of which were published from 2013 onward. All relevant information of these articles is available in the Appendix A. 

The hybrid coding of the 15 synthesis memos resulted in nine primary codes and 15 secondary codes, outlined in Table 1 along with the full codebook available in the Appendix A. Upon ATA, four main analytical themes surfaced: (a) support preferences; (b) challenges to suicide programming; (c) integration of culture as prevention, and (d) grounding relationship in prevention. Below is the expansion of each of these four themes and respective subthemes.

### 3.1. Theme 1: Support Preferences 

The literature emphasized the need for social and behavioral health supports when treating or promoting suicide prevention practice. The dichotomy that lies between formal and informal supports was evident with respect to both the positive components and negative repercussions in the use of either.

#### 3.1.1. Formal Supports 

Formal services are broadly referred to as behavioral health services provided by licensed professionals or unlicensed paraprofessionals operating within a non-profit, government agency, or private practice. Community services and other paraprofessional services such as school counselors, health and human services (e.g., case managers), and suicide prevention hotlines, such as the National Suicide Prevention hotlines or the Native Youth Crisis Hotline are also considered formal supports. Youth and young adult participants recounted histories of seeking out formal supports and noted that the care they received was helpful as it provided them with safe space to share and reflect on past traumatic events and to develop coping strategies. Overall, preference for utilization of such formal supports is low. Burrage and colleagues (2016) explain the implications when western constructs of treatment are hailed over Indigenous cultures of healing:


*…when respecting the assumption that traditional AI culture both promotes a healthy lifestyle and aids in healing, one might view the clinical conceptualization of suicide prevention as forming part of a mainstream culture that is in itself unhealthy for AI youth. If prevention is viewed only as increasing access to clinical services and educating community members about suicide, it leaves little room for prevention strategies that are more culturally appropriate.*
[42] (p. 146)

This perspective challenges the centralization of clinical services as the basis for suicide programming and addresses the negative connotations associated with the promotion and revitalization of cultural and traditional health practices held in clinical settings. Trout and colleagues (2018), too, note that “*by professionalizing suicide care, the assessment and management of the suicidal mind becomes a responsibility of the select few, and the broader contexts of community and cultural support fade from view*” [42] (p. 402). 

#### 3.1.2. Informal Supports 

Informal supports are identified as family, including parents or siblings as well as cousins, elders, and other trusted individuals in the community, friends, elders, youth leaders, mentors, traditional practitioners, or individuals that are identified as being in alignment with an individuals or communities’ values, belief systems, and practices. The inclusion of family, friends, and other informal supports impacted participants’ decisions to seek formal support by increasing uptake. Individuals that offer aid similar to that of professionals can help navigate resources and barriers, as well as social and cultural support. Traditional practitioners are often designated as informal supports although Burrage and colleagues (2016) explain that:


*Although traditional healers are generally categorized as informal supports, such definitions privilege dominant understandings of mental health and wellbeing. Traditional healers may not be professionally trained in the provision of services, but in many cases their training and experience goes beyond that of trained professionals.*
[42] (p. 144)

The US healthcare system, framed within settler-colonial logics does not integrate multiple truths or ways of knowing. There is an opportunity to shift mindsets and understandings of wellness via traditional pathways to healing in conjunction with the dismantling of privileged status of clinical practice to evolve outside of physical ailments and holistic approaches to care. 

#### 3.1.3. Environment 

One’s home environment influences suicide risk and utilization of services. Shaw et al. (2019) reported that participants had experienced sexual abuse as a child and into adulthood and could recall first memories tied to abuse in their homes as early as 3 or 4 years of age. Some individuals who had grown up around or within environments of chronic alcohol and substance misuse, had misused or are now addicted themselves. Despite these challenges, many youth and young adults cannot leave their home environments safely, or an individual in a domestic abuse situation may not easily be able to remove themselves from their environments [43]. 

Additionally, the economic climate impacts one’s environment, employment, and training opportunities conducive to wellbeing. de Schweinitz et al. (2017) shared participant views on the difficulties of securing paid work. They highlight that the inability to secure work for stability and other resources (e.g., healthcare insurance) often tied to employment contributes to suicide [44]. Skewes et al. (2022) too described how an increase in economic opportunities can positively influence the individual, familial and community levels—even when such opportunities are at the expense of being away from one’s tribal community or exposes individuals to structural racism and bias [45]. 

### 3.2. Theme 2: Challenges to Suicide Programming

Across the sample, challenges to conducting suicide programming were identified. The foundations of these challenges are linked to historical traumas leading to barriers to care, the need to maintain self-reliance, and issues with generalizability as best-practice. 

#### 3.2.1. Historical Trauma 

The impact of historical memory of colonization and continued forced assimilation is set as the foundation for suicide programming. Troute et al. (2018) shares that, “historical memory of the impacts of colonization…the impact of boarding schools, the loss of language and traditional ways, challenges posed by living “between two worlds”, and struggles with the current school system that continues to marginalize traditional ways. [46] (p. 401). A participant continued to specify that acknowledging the ways in which historical trauma has impacted suicide is what “…help[s] us understand why we are having these problems like we’re experiencing now and how we’re going to address it” [46] (p. 400). Participants across the literature addressed and shared personal stories of the harms of historical trauma on the family system—which in turn have impacted relationships and social connection that combat suicide risk. Amidst continued marginalization, Indigenous communities foster opportunities for resiliency and nourishment, especially within the family system. Strickland et al. (2006) describes historical trauma as tied to the loss of traditions and the pathways to ensuring the continued sacredness of familial relationships:


*…holding the family together and the culture; in this respect, they worried about potential loss of traditions and the fracturing of families from historical trauma and the effect of modern day life…All participants voiced concerns about the family and the effect of modern day life on family values. Holding the family together meant assuring communications across generations, passing on traditional parenting skills and teachings, providing support holistically…*
[47] (p. 8)

The connection between family systems and culture influences the impact of historical trauma at all socio-ecological levels. The positive counteraction to the impacts of historical trauma inflicted upon Indigenous Peoples with that of relationships across the generations is profound. 

#### 3.2.2. Barriers to Care 

A lack of general knowledge among the community regarding suicide prevention is common barrier to care. Participants shared that a lack of educational materials and resources regarding suicide and suicide prevention leaves community members unable to identify the signs or intervention strategies to support those in need [42]. Relatedly, the tendency for people in the community to stay reserved in sharing personal issues of suicidality or reluctance in discussing behavioral health concerns in general is another consistent barrier to care. Rasmus et al. (2014) shares the voices of a working group in a rural AN village explaining that the practice of speaking openly about such topics was not common, but rather just keeping things to oneself, not just regarding suicidality, but within the Yup’ik traditional context, the norm was to avoid talking about any negative feelings (e.g., anger, frustrations). The elders spoke of their experiences in not being able to discuss such things openly as the power of words transforms, meaning the thought of speaking of the experiences would bring them into existence. These feelings though could be expressed in other settings and to other relatives, such as animals, rocks, plants, and being in natural environments that foster reflection through harvesting and food gathering to release negative feelings and stressors [48]. With respect to the urban Indigenous experience, Burrage et al. (2016) outlined participants reflections on the challenges related to accessing services and care. The urban population has been viewed as invisible, having individuals being misidentified ethnically or experiencing interactions that perpetuate the stereotype that Indigenous Peoples no longer exist. Participants communicated barriers to transportation and reliable public options for youth and young adults that are not dependent on permission from adult guardians. The overall lack of availability for health care and behavioral health professionals was identified as a barrier as well as the lack of investment in Indigenous communities from government entities with power to instill change. Moreover, the inability or barrier in paying for services and the urban Indian organizations that may provide such services are also operating with limited financial resources [42]. 

#### 3.2.3. Self-Reliance 

It was a common expression among participants that they should deal with their experiences and be self-reliant without impacting others with the request for support. Participants acknowledged that everyone has their own problems and experience and did not want to burden others with their own [49]. Part of self-reliance in keeping a balance of a positive mentality and positive behaviors are activities such as hobbies, exercise, spiritual or cultural practices. In addition, many individuals thought of how their suicide would impact their loved ones and the harm that suicide can bring to a community. Participants noted that being held accountable to themselves and to their communities by sharing their story and experiences could support those who may be suicidal, as well as supporting themselves in their ideation. Shaw et al. (2019) wrote that self-reliance came about more prominently when participants thought of their death by suicide impacting their loved one. The harm that they would cause was what increased awareness and acknowledgment of their suicide ideation and accountability to themselves. Participants expressed that sharing their own experiences with others was a way to help themselves heal as opposed to seeking out formal supports. Many reflected that by being a survivor of suicidality brought back a sense of purpose and future orientation for them and the safety of their communities through the sharing of their stories [43]. 

#### 3.2.4. Generalizability 

In these intimate programs and interventions, the sharing of one’s experiences is a gift, a gift of time and wisdom as it was asked of participants to share pieces of themselves and that of their respective communities. Such gifts cannot be ‘plugged and chugged’ into program development but rather woven into the fabric of the efforts themselves:


*You’ve just sat with somebody for 2 h and you invest incredibly personal information and made them go deep into their own self…You’ve been given the privilege and the honor to hear this information. You need to respect that and thank that person for that and continue that connection. If you go from that to a standardized something, you’ve discounted what you just had I think.*
[50] (p. 8)

Jansen et al. (2021) reflected upon their findings to *“…challenge the notion of a single treatment for all groups of people. For AI/AN populations, a collaboratively designed, community-based approach grounded on local knowledge is best practice*” [50] (p. 14). There is a need for cultural adaptations and culturally grounded interventions that align with the values, traditions and voice of the Indigenous population in collaboration. ‘Pan-Indian’ approaches to developing programming with a focus on replication and transferability goes against the main principles of culturally centered efforts by which the program itself is designed to reflect and support needs identified by the community. 

### 3.3. Theme 3: Integration of Culture as Prevention 

Participants across the literature shared commonalities between Indigenous communities that share values and beliefs in providing mutual support, characteristics of helpers and healers, as well as the concern surrounding the erasure of cultural identity and to integrate traditional healing in preventative efforts. It is a shared and community responsibility to keep everyone safe, healthy and whole through community resilience, kinship traditions, traditional pathways for intergenerational knowledge exchange, and sovereignty. 

#### 3.3.1. Community Resilience

Community support systems are preferred to professional and formal supports, which are often delegated or conducted by non-Indigenous practitioners. Although resources are limited, it was shared that community members would likely share money if they can support local resources and education or even gift their time through volunteering to assist in suicide preventions efforts. A participant emphasized that the health and human services provided to federally recognized and tribal communities managed and distributed by the federal government was not without cost, nor has it been thoughtfully appropriated [47]. Community supports can help youth feel comfortable and understood and the presence of culture, mutual support, and aid is valuable. The loss of cultural identity among the urban AI community as a result of settler colonialism is imperative to address. Participants across the sample share how the space and opportunity to share their perspectives regarding suicide and its impact on their communities was an act of healing and transformation. Moreover, DeCou and colleagues (2013) found that the outcomes of the interviews encouraged participants to grow their understanding of suicide: 


*…some participants expressed the realization of their own voices in contributing to the ongoing discussion of suicide and suicide prevention…In finding their voice this participant echoes the outcome experienced by several participants, whereby the interview experience allowed for the reinterpretation of previously held beliefs.*
[51] (pp. 73–74)

Participants detailed the need for traditional healing practices and practitioners to teach the community, for shared knowledge systems, to better understand themselves to serve others, and the use of storytelling practices to strengthen community resilience as suicide prevention [42,47]. 

#### 3.3.2. Kinship Traditions 

One’s cultural identity is influenced by their respective kinship connections within community as well as access to traditional and cultural knowledge. Kinship systems encompass family systems and how family members are identified and determined based on relation, which also translates to how children are raised and developed through a practice of community caregiving. Indigenous caregiving challenges western convention of family systems since it is not exclusive to biological relation. This is imperative to note, as relationship and communication with one’s kin was shared as a protective factor against suicide [47,48]. Cultural practices are carried out in a community with others strengthening kinship systems as well as cultural identity. For example, the gifting of ‘Indian names’ was addressed as a large component of one’s tribal identity [52]. Rasmus et al. (2014) described the generational change between naming practices as represented by Elders and that of the youth today. Young people’s first names may lack the connection and purpose behind their name that their Indigenous name could provide. First names may not be closely tied to one’s cultural identity or to that of their familial kin with respect to roles, responsibilities and sharing of knowledge systems [48]. This change in generational understanding of names and kinship traditions has an impact on community connection. Revitalizing traditional lifeways that sustain kinship value systems is instrumental to individual, familial, and community level wellbeing as the practice hones in on relationships. 

The promotion and integration of culturally grounded interventions at the familial and community level are needed to disrupt suicide risk as well as challenge larger systems of oppression and discrimination that contribute to risk [47]. Such interventions and approaches can include “*…cultural activities and traditions—for example, subsistence activities, cultural dancing, and potlatches—may promote multiple overlapping experiences of interpersonal connection and contribute to a protective sense of belonging and positive cultural identity*” [45] (p. 85). All of which strengthen the community rather than solely at the individual level. Traditional practices teach us that we are not independent of other beings, but rather connected. Understanding suicide risk and protective factors through a familial and kinship lens offers new insights and approaches to prevention as well as treatment. 

#### 3.3.3. Traditional and Intergenerational Knowledge 

Across the sample it was evident that knowledge sharing is imperative to cultural revitalization and sustainability across generations. Participants call for and encourage the development of strategies that *“…bridge the gaps between youth and the older generations. Touching on the issue of intergenerational mentorship and support*” [46] (p. 401). Such mentorship can encompass traditional healing practices that integrate and acknowledge holistic health and wellbeing practices across Indigenous communities. Traditional knowledge specific to food, gathering, and sustenance as well as knowing one’s native language are important because they nurture physical bodies, the community, and relationship with the natural world. To share one’s culture and teachings to the next generation, is to express love and mutual respect for the passing and evolution of traditions. Rasmus (2014) highlighted Elder perspectives in the process of prevention programming and the leadership and guidance of Elders to initiate program development and implementation [53]. One Elder specifically shared how: 


*The Elders and young people meeting together is the best part of this program…we start telling about the right way to do things and it start opening things up—and it’s bringing out the Elders in a positive way. We saw a vision. We want to bring people together. And young people talk about the meetings and ask—when are we going to meet again. It’s like a net we have neglected—if it needs mending to fix the holes. This project is like a big net—we are catching people here in the community with this net—dead or alive and if dead they might come alive again…It’s time to wake up. It’s time to come alive.*
[53] (p. 174)

The metaphor of the net shares how the community ‘catches’ and supports people when they are in need and collectively aids individuals who may be struggling. The focus of the Elder driven suicide prevention program [52] was led and facilitated by Elders directly influencing relationship capacity and connection with youth. Healing and coping strategies are drawn from traditional practices and the process of teaching that intergenerationally greatly influences wellbeing: “*I was taught by the elders that when you are blue and sad to go to the river and let the river draw that sadness out of you*” [47] (p. 9). The focus on sharing knowledge across generations emphasizes unity in cultural protocols and traditional medicine practices that foster strong relationships as well as community caregiving [45]. 

#### 3.3.4. Sovereignty 

Authors outlined participants’ perspectives regarding sovereignty and the rights of decision making within their respective Indigenous communities. This is especially relevant related to traditional knowledge and medicine, social systems, and how protocols, interventions and care are conducted—leadership and direction from Indigenous communities should drive prevention and treatment services [46]. The assumption that western approaches are above that of Indigenous knowledge and ways of knowing was viewed as negative by participants and, in some ways, continues to force assimilation and further condones stigma and risk for behavioral health outcomes such as suicide. Hence, sovereign nations and Indigenous leaders are standing strong in their power to ensure that the transformations, changes, and data collected within their communities are not only well represented but ensure the safety of their people and their knowledge from further exploitation or negative experiences of health care. Cwick et al., (2019) shared a note to their readers as to why there were no direct quotes of qualitative analysis within their article, *“(Note: The Elders wanted these interviews to only be used to develop the curriculum themes, not to be shared in the public domain as part of research)”* [54] (p. 140). 

### 3.4. Theme 4: Grounding Relationship in Prevention 

Across the sample there was a focus on relationships as the foundation for prevention strategies; as a counternarrative to dominant western ideology that centralizes outsider researchers as sole knowledge bearers. Highlighting relationship and community collaboration is key for the development, implementation, continuation, and effectiveness of suicide prevention programming with Indigenous Peoples. Communities are responsible for the wellness of one another through community caregiving, knowledge sharing, and reciprocal collaborations. These ensure the strength and consistency of accountable communities, academic partnerships, if and when appropriate, as well as the sustainability of prevention efforts. 

#### 3.4.1. Accountable Communities 

Participants emphasized the need for widely available education regarding suicide and behavioral health that come from trusted community members and connects to their ways of knowing and cultural considerations. Participants highlighted that the sharing of stories and experiences related to suicide through peer mentorship and advocacy encourages conversation and community collective voice to foster change and implement action. The sharing of stories can encourage the healing and awareness of others within the community. Connection and bonds were fostered out of these spaces of oral storytelling and qualitative inquiry, encouraging community members to maintain relationships outside of the ‘research’ space [48,51,55]. As stories are shared and emotions are expressed, it was noted by participants that it is imperative to uphold confidentiality. While sharing one’s story can uplift another, names or other identifying information within small communities are important to stay anonymous, especially within rural or smaller tribal communities. The risk of confidentiality being broken fuels apprehension toward seeking or accepting help when they need it most [51]. 

Moreover, community-specific initiatives need to be led by community members. As a prime example, Antonio and colleagues (2020) outlined how youth leaders shared that Hawai‘i’s Caring Communities Initiative elevated their skills related to suicide prevention and leadership because the program itself was youth driven [55]. All authors throughout the sample identify community mobilization as an important factor in engaging the community and identifying capacity to support prevention efforts for suicide. Cwik and colleagues (2019) highlighted that the Elder council developed materials that ensured that the intervention curriculum was built from the ground up to reflect and integrate the unique culture and values of the Apache, as well as to highlight traditional oral storytelling practices. This inclusion of the practice of respect is the basis for relationships that support trust and change as driven by the community for the community [54]. The weaving of respect for one another promotes learning, accountability, and mutual aid amongst Indigenous communities, all of which can promote and encourage suicide prevention efforts. 

#### 3.4.2. Academic Partnerships 

Integrating and collecting community member perspectives as a part of the experience and process of intervention development is vital. This counteracts the common experience of tribal communities where the researchers at university come from the outside and impose their systems of knowledge onto the tribal community rather than collaborating alongside them. As part of community involvement, there was an emphasis across the sample to include the wisdom and perspectives of Elders, leaders, youth, and community members through advisory boards, committees, as facilitators, and program participates. As outlined previously, these contributors play an essential role in the cultural strength and resiliency of the tribal community and helps to build community capacity and leadership in traditional knowledge systems. Rasmus (2014) noted that the intergenerational knowledge exchange from Elders in partnership with the university research team led to reciprocity of learning—the research team found themselves immersed and encultured in the Yup’ik ways and more connected with the community over time. Participants impress upon university partners that the research and programming developed overtime does not belong to the university [53]. Rights and ownership of knowledge are not to be challenged and the sharing of collaborative work must be carried out ethically and inclusive of all voices, especially those of the community [55]. 

#### 3.4.3. Sustainability 

All authors mention in some capacity the tendency for prevention interventions within tribal communities to end once university partners complete their deliverables and the finances brought with them are exhausted. Per Antonio and colleagues (2020), even the funding allocated to conduct the program per the set timeline did not allot enough time to develop relationships that could sustain efforts, nor to support youth leaders’ goals. Community coordinators highlighted a lack of organizational support to facilitate increased buy-in and more leaders to continue to commit to the program. Increased relationships to bring in organizational resource support were needed to extend efforts beyond the allotted programmatic timeline [55].

Overwhelmingly, the preference for community-based participatory action approaches were noted to yield better programming and initiatives for Indigenous communities, however authors shed light on the impact and asks of the community that such approaches require. Rasmus (2014) notes that although preferred and best-practice for Indigenous Peoples, they are time consuming and ask much of the community and can result in retention issues as well as decreased community involvement overtime, especially when their time is volunteered or gifted, specific cultural activities have subsided due to the seasons, or suicide-related incidences have decreased [55]. Community ownership and buy-in require consistent relationship building and a reassessing of capacity throughout a programmatic timeline. Ensuring community voice is heard and appropriate changes are made can, in some cases, increase involvement. Le and Gobert (2015) acknowledge that university or academic protocols for meetings or benchmarks may not serve the community, suggesting the need for more creative and intuitive approaches to ensure program milestones are met, while also building upon relationships and sustainability efforts to cultivate proactive suicide prevention initiatives [56]. 

## 4. Discussion

The current meta-synthesis was conducted to examine how participant perspectives were integrated via qualitative data to inform program development to prevent suicidality amongst Indigenous Peoples, to identify the ways in which participants stories and narratives were included, and to report on thematic patterns and contradictions that emerged throughout the analysis. In conducting ATA, the following themes surfaced, all of which were identified as interconnected with one another: support preferences, challenges to suicide programming, integration of culture as prevention, and grounding relationship in prevention. Each of these four themes had respective subthemes outlined to further describe the patterns and contradictions that emerged. 

The first theme, support preferences, addressed the limited availability of safe spaces, preference for ‘informal support’ such as family or peers over ‘formal’ services such as support groups and counseling, as well as barriers to care inclusive of cost and transportation for rural participants, triaging, and care for urban participants as well as culturally integrative or holistic care with access to traditional medicine practices. The use of ‘informal’ and ‘formal’ language was further challenged by Trout and colleagues (2018), noting that the professionalization of suicide care within the western healthcare model further perpetuates barriers in accessing care and further diminishes traditional healing practices of respective Indigenous communities. Similarly, participants called out the need for coordinated care from practitioners who are Indigenous to provide care that extends beyond the individual to the family and community [48]. 

The second theme, challenges to suicide prevention, noted the impact of historical trauma in respective Indigenous communities today. Mitigating further acculturation and barriers to access to care involves acknowledging the relationship between the factors of historical trauma and the impacts of colonization on suicide risk. These relationships must be continually investigated, experiences must be shared, and voices heard to shape culturally informed prevention strategies [57]. Of note, these barriers as tied to systemic social injustice have led to individual’s apprehensive to asking for help or to misinterpret care when it is received and feeling dependent on others, or a burden [55]. Generalizability is promoted and expected by much western academic practice which has led to the perpetuation of Indigenous seen as a monolith or fostering ‘Pan-Indian’ approaches that do not respect nor highlight the unique cultures, languages, or characteristics of respective Indigenous communities [50]. 

The third theme, integration of culture as prevention, revealed commonalities in cultural practices that are woven across Indigenous communities, while still speaking to the unique and diverse approaches to care found among them. Specifically, the importance of intergenerational knowledge exchange, family systems and cultural identity that are integral components to one’s healing and wellbeing. All of which are congruent with the current literature on the integration of culture in prevention and treatment [23,58,59]. Rasmus and colleagues’ (2014) challenging position on ‘culture as prevention’ approaches are notable [48]. Their words aim to broaden awareness and acknowledgement on how such approaches can further promote false narratives of pre-colonial times by focusing in on some practices and protocols that may perpetuate harm, which can contribute to individual and collective traumas [60]. 

The fourth theme, grounding relationship in prevention, centers relationships as the basis for effective and sustainable suicide prevention strategies. A community of care is emphasized as a traditional and current practice that ensures accountable practices in sharing stories vulnerably and accessing care as well as the role of the community buy-in and ownership of wellness initiatives encompassing suicide prevention. Academic partnerships, with community engaging with universities to conduct research projects, elicits transparent discussions surrounding ownership and acknowledgment of sovereignty. Le and Gobert (2015) speaks to the rights and standing of respective Indigenous communities to determine whether they will continue with a cultural program [56]. Although revered as best-practice amongst Indigenous Peoples, academic partnership shortcomings include an extended timeline of multiple years, community burn-out, reduce momentum, and can limit funding and sustainability [48,55]. The long timeline is warranted to build relationship and to develop culturally grounded interventions; however, when the project timeline ends, often so does funding. 

The relationship between historical and perpetual trauma resulting from colonization must be acknowledged and investigated; experiences must be shared, and voices heard to shape culturally informed prevention strategies. This relationship is central to understanding and highlighting appropriate and integrative approaches to suicide prevention. When woven with culturally grounded approaches and reciprocal collaborations, prevention programs can support protective factors such as reclamation of cultural identity, spirituality, positively reinforced emotion regulation, cultural continuity in community relations, trusted communication, and increased social connection [58,61]. Suicide prevention programs that work alongside Indigenous Peoples can address said factors as behavioral health access and social connectedness have been shown to increase protective factors [62]. Although the existing literature is sparse, the research that has been carried out shows that a protective factor-focused, strengths-based approach is most effective in addressing suicidality throughout Indigenous communities [63]. Garroutte and colleagues found through qualitative analyses that a “*commitment to cultural spirituality, as measured by an index of spiritual orientations, was significantly associated with a reduction in attempted suicide*” [59] (p. 1571). Furthermore, research conducted with tribal nations in the Midwest and Northern Plains has similarly found that connection to land, place, and cultural identity aids individuals and communities alike to navigate barriers and stressors, which leads to a decreased risk for suicide [64]. 

Efforts that acknowledge and address impacts of historical trauma on suicidality and focus on Indigenous communities’ inherent strengths have been proven to be more effective and provide culturally integrative strategies. These strategies are more effective than current deficit models that are often racialized, centering whiteness, and failing to engage and serve Indigenous Peoples [65,66]. Therefore, prevention and intervention programs that center reciprocal collaborations between the researchers and the population must be prioritized with Indigenous Peoples [67]. Approaches such as community based participatory research (CBPR) and participatory action research (PAR) strategies focus on conducting projects alongside and in partnership with oppressed, underserved, and marginalized populations such as Indigenous Peoples [27,29,68]. 

### 4.1. Limitations 

The limitations of this meta-synthesis are due in part to the methodological approach. Because the focus was to analyze peer-reviewed suicide interventions and programs, specifically those that included qualitative methods, only 15 articles conducted with Indigenous communities were identified. It was beyond the scope and practice of this meta-synthesis to examine the results of the interventions to compare quantitative metrics of impact or to assess differing perspectives and experiences of suicidality based on biological sex. Thus, interventions were not evaluated in this way. Moreover, the focus on peer-reviewed programming leaves the possible exclusion of culturally grounded programs that exist in communities that were not captured in this meta-synthesis. In addition, the linguistic limitations of those articles published in English, leaves the possibility of other relevant articles and written pieces published outside of scholarly journals and those written in other languages. 

### 4.2. Implications 

Future research and programming should invest in the de-stigmatization of suicide to foster a greater understanding of suicidality and to promote increased access to behavioral health care. Interventions that support community education and coping strategies will promote resiliency and position physical, social, and emotional wellbeing in balance to create meaning and influence change. The findings of this meta-synthesis emphasize the need for health systems, crisis hotlines, providers, allies, and collaborators to work alongside Indigenous communities to decrease risk for suicide and foster a heightened awareness of how suicide impacts other socio-ecological levels, such as the family and community, as well as increased social support, belonging and connectedness [43]. This meta-synthesis was carried out through review of research and programming conducted alongside Indigenous Peoples, with several distinct Tribal communities represented. While Indigenous Peoples are diverse in many ways, the perspectives from a single Tribal community do not represent all Tribes, there is value in centering Indigenous perspectives in suicide prevention outside of the US, as values surrounding cultural traditions, protocols, and impacts of historical trauma due to settler-colonialism discussed are applicable to the broader goal of advancing suicide prevention strategies among Indigenous populations. However, we acknowledge and do not assume that these are generalizable to all Indigenous communities. These efforts highlight the importance of community-driven and culturally grounded prevention programming. 

The lack of community-engaged qualitative methodological approaches within the literature to support program development is evident. There is opportunity to further guide qualitative works in prevention practice through the lens of critical and Indigenous methodologies [69]. Findings suggest that the implementation and sustainability of prevention efforts are dependent on the engagement and voices of Indigenous communities, reciprocal academic and healthcare system collaborations, as well as sustainability strategies that support the financial and capacity security beyond designated timelines, often determined by granters. Such efforts require cultural humility, communication, and trust. The use of action research to inform culturally sound suicide prevention programming should be applied when designing, implementing, and evaluating strength-based strategies [47]. The current literature predominantly focuses on the social and cultural analyses, leaving a gap in economic research regarding help-seeking that encapsulate finances, biological sex and gender identity, and decision making. More in-depth investigations beyond the empirical literature and western strategies are needed to hear and learn from the experiences of Indigenous Peoples, in addition to that of their social supports and healthcare access preferences to uncover cultural nuances and structural needs to prevent suicidality [49]. 

## 5. Conclusions

To our knowledge, this is the first meta-synthesis to assess suicide prevention interventions and programs amongst Indigenous Peoples, grounded within an Indigenous Ways of Knowing theoretical framework. In alignment with cultural praxis, data were interpreted through a cultural context [70] in an attempt to widen the understanding of approaches to suicidality across Indian Country. The current study provides further insight into the heterogeneity of Indigenous populations, to hold space for Indigenous voices to be further centralized in the realm of prevention science. That centralization bears the promise of elevating Indigenous voices to improve the very programs that exist to serve them. 

## Figures and Tables

**Figure 1 ijerph-20-07064-f001:**
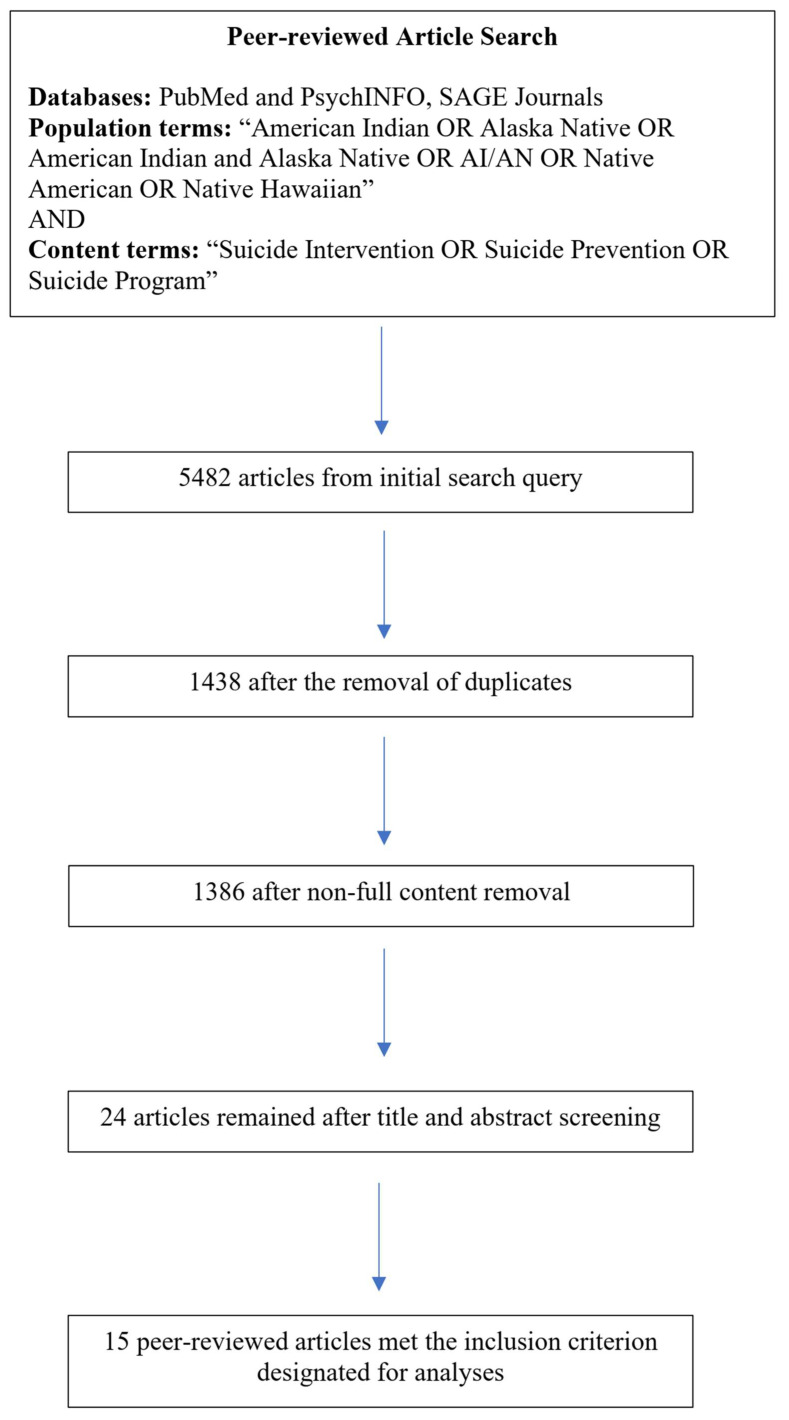
Flowchart of article selection process.

**Table 1 ijerph-20-07064-t001:** Primary themes and respective subthemes.

3.1. Support Preferences	3.2.Challenges to Suicide Programming	3.3. Integration of Culture as Prevention	3.4. Grounding Relationship in Prevention
3.1.1 Formal Supports 3.1.2. Informal Supports 3.1.3. Environment	3.2.1. Historical Trauma 3.2.2. Barriers to Care 3.2.3. Self-reliance 3.2.4. Generalizability	3.33.1. Community Resilience 3.3.2. Kinship Traditions 3.3.3 Traditional and Intergenerational Knowledge 3.3.4. Sovereignty	3.4.1. Accountable Communities 3.4.2. Academic Partnerships 3.4.3. Sustainability

Primary themes are highlighted in the gray, with secondary themes outlined below them.

## Data Availability

Data for this meta-synthesis were the final articles in the sample determined by the filtration process (*N* = 15). Synthesis memos were created for each article and then analyzed. Details on these data are available in the Appendix A.

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
