# Peer review of "Indigenous Voices Against Suicide: A Meta-Synthesis Advancing Prevention Strategies"

_ijerph, 2023, doi:10.3390/ijerph20227064_

Round 1

Reviewer 1 Report

Comments and Suggestions for Authors

I would like to thank the academic editor for the invitation to review the paper titled “Indigenous voices against suicide: A meta-synthesis advancing prevention strategies.” The authors discuss the critical importance of centering Indigenous voices and perspectives in the development and evaluation of suicide prevention programs for Indigenous Peoples of the United States (IPoUS). The meta-synthesis of qualitative research sheds light on the current state of suicide prevention efforts, revealing a lack of participant-centered approaches. This work holds significant implications for advancing culturally adapted suicide prevention and promoting strategies rooted in cultural understanding and sensitivity.

Introduction

Overall, the introduction effectively establishes the gravity of the issue and sets the stage for the study's objectives. However, I personally found it to be very lengthy, and I believe a plethora of information may not be required. The introduction should probably be limited to 1-2 pages instead of 4 pages. I believe this would improve the overall readability of the paper. For example, the theoretical approach adopted in the paper might be something that could fit into the methods section. A lot of information and comparison narrative from 'Impacts of Colonization on Suicide Risk' and 'Cultural & Community Integration Within Suicide Prevention Programming' may go into discussion.

While you've highlighted the need for culturally-informed suicide prevention, you could explicitly mention how your study contributes to this field. What specific insights or new perspectives does your research offer? This can make your study's significance even more apparent.

Methods

It would be beneficial to mention any specific search terms or combinations that were used. This would add transparency to your search process and help readers understand how you identified relevant studies. This is important for replicability. I recommend including the detailed search strategy for at least one database as an appendix.

The explanation of the study selection process is clear and well-structured. However, consider providing a bit more context on how many articles were identified at each stage. This can give the reader a sense of the scale of the initial search and how it was narrowed down.

The author had 24 articles after screening and included 15. I believe that for those 9 articles, you can provide a supplementary section to show why each article was removed from the final synthesis.

Was coding verified by both authors? I believe the author may need to add details about the rigors associated with data extraction. How were errors identified?

Results

Well drafted, however, the results could have been presented more succinctly. I advise revisiting the results section and trimming it down. The readability of the results is poor.

Consider moving Table 1 to supplementary material.

I highly recommend reducing the word count and also presenting the themes and subthemes in a tabular or graphic format. This would greatly enhance the readability of the paper. Readers might lose track/ flow while reading, so having a pictorial roadmap of themes would be beneficial.

Discussion

Well drafted.
I do believe that the findings from the study's application are not limited to the US but also relevant to other native populations worldwide, possibly providing a foundational approach. I think some relevance to a global scale can be added. For example, does your study have any implications for Aboriginal populations in Australia, native populations in India, etc.? IJERPH has a global readership, so why do you want to publish the paper in an international journal rather than a national one? Because it does have implications for global academics and researchers.

Comments on the Quality of English Language

None.

Author Response

Thank you for the feedback regarding our manuscript Indigenous voices against suicide: A meta-synthesis advancing prevention strategies. We are very appreciative of the depth of engagement with our work and the feedback provided.  

We have responded to each of the comments, including a description of the content and location of changes in the revised manuscript. You will also see that we made some changes in the terminology used to better represent Indigenous populations in the United States, as ‘Indigenous’ or ‘Indigenous Peoples’, rather than Indigenous Peoples of the United States (IPoUS), per the guidance and recommendation of Indigenous scholars, mentors and peers. 

Introduction 

1. Overall, the introduction effectively establishes the gravity of the issue and sets the stage for the study's objectives. However, I personally found it to be very lengthy, and I believe a plethora of information may not be required. The introduction should probably be limited to 1-2 pages instead of 4 pages. I believe this would improve the overall readability of the paper. For example, the theoretical approach adopted in the paper might be something that could fit into the methods section. A lot of information and comparison narrative from 'Impacts of Colonization on Suicide Risk' and 'Cultural & Community Integration Within Suicide Prevention Programming' may go into discussion. 

We appreciate the reviewer's suggestions and have reorganized accordingly. We have moved the Theoretical Framework section to the methods followed by Author Positionality starting at line 109. Some of the narrative from the ‘Impacts of Colonization on Suicide Risk’ and the ‘Cultural & Community Integration within Suicide Prevention Programming’, although this subheading has been removed and the narrative from this previous section has been transferred into the Discussion section starting at line 670   

2. While you've highlighted the need for culturally-informed suicide prevention, you could explicitly mention how your study contributes to this field. What specific insights or new perspectives does your research offer? This can make your study's significance even more apparent. 

Language has been modified to highlight this study’s significance. You will find these on lines 94-97 as well as line 101.  

Methods 

3. It would be beneficial to mention any specific search terms or combinations that were used. This would add transparency to your search process and help readers understand how you identified relevant studies. This is important for replicability. I recommend including the detailed search strategy for at least one database as an appendix. 

We thank the reviewer for the opportunity to further clarify and offer insights into the screening protocol which can be found in the supplementary materials. Within the protocol search content terms and search string combinations are identified.  

4. The explanation of the study selection process is clear and well-structured. However, consider providing a bit more context on how many articles were identified at each stage. This can give the reader a sense of the scale of the initial search and how it was narrowed down. 

Details regarding the screening protocol and study selection are outlined in the screening protocol available in the supplementary materials.  

5. The author had 24 articles after screening and included 15. I believe that for those 9 articles, you can provide a supplementary section to show why each article was removed from the final synthesis. 

We appreciate the opportunity to further clarify and provide insights into the screening protocol. In alignment with transparency and replicability, we have included the search strategy and protocol which includes the reasoning for removal of those 9 articles in supplementary materials. The protocol highlights the filtration process for each database, resulting in the 15 articles included in the final synthesis.  

6. Was coding verified by both authors? I believe the author may need to add details about the rigors associated with data extraction. How were errors identified? 

Many thanks for the opportunity to expand on coding verification and rigor of review filtration. Two coders, the first author and a research assistant (noted in the ‘acknowledgments’ section of the manuscript) were involved in the filtration process, as noted in lines 178-184. Conferencing meetings occurred between coders to discuss rationale and reach consensus. The second author was designated as a conflict resolver, if the two coders came to a disagreement. However, conflict resolution was not needed at any point throughout the analysis. Per the guidance of a qualitative expert and mentor (noted in the ‘acknowledgments’ section of the manuscript), considering the small sample size of 15 articles, the synthesis memo development, coding and applied thematic analysis of the memos was conducted by the first author and reviewed by the second author for verification.   

Results 

7. Well drafted, however, the results could have been presented more succinctly. I advise revisiting the results section and trimming it down. The readability of the results is poor. 

We have heavily edited the results section, reducing the presentation of each theme and subtheme to about one paragraph and presenting only the most salient block quotes. This reduced the results section, which was originally 12 pages of text plus the table, to seven pages of text plus a table. 

8. Consider moving Table 1 to supplementary material. 

Per the reviewer's recommendation, ‘Table 1’ has been removed from the manuscript, and relevant information regarding the articles has been moved into supplementary materials and is noted as such on line 237 and 768.  

9. I highly recommend reducing the word count and also presenting the themes and subthemes in a tabular or graphic format. This would greatly enhance the readability of the paper. Readers might lose track/flow while reading, so having a pictorial roadmap of themes would be beneficial. 

Many thanks for the reviewer’s suggestion to outline the themes and subthemes in a visual format. You will find that an additional Table 1 has been included starting at line 245. The full codebook has also been included as part of the supplementary materials.  

Discussion 

Well drafted. 

10. I do believe that the findings from the study's application are not limited to the US but also relevant to other native populations worldwide, possibly providing a foundational approach. I think some relevance to a global scale can be added. For example, does your study have any implications for Aboriginal populations in Australia, native populations in India, etc.? IJERPH has a global readership, so why do you want to publish the paper in an international journal rather than a national one? Because it does have implications for global academics and researchers. 

We appreciate the reviewers' note that the discussion section has been drafted well. We chose to submit to IJERPH specifically for the special issue entitled "Advances in Indigenous and American Indian and Alaska Native Health and Wellness." Although, we agree that implications of this study may share relevance to other Indigenous populations, we do not wish to over generalize applicability considering the differing historical, political and social factors experiences by Indigenous Peoples internationally. We have added language reflecting this in lines 722-730, and reads as follows:  

“This meta-synthesis was carried out through review of research and programming conducted alongside Indigenous Peoples, with several distinct Tribal communities represented. While Indigenous Peoples are diverse in many ways, the perspectives from a single Tribal community do not represent all Tribes, there is value in centering Indigenous perspectives in suicide prevention outside of the US, as values surrounding cultural traditions, protocols, and impacts of historical trauma due to settler-colonialism discussed are applicable to the broader goal of advancing suicide prevention strategies among Indigenous populations. However, we acknowledge and do not assume that these are generalizable to all Indigenous communities”.  

Reviewer 2 Report

Comments and Suggestions for Authors

I would like to thank the authors for carrying out such a relevant and interesting work. Suicidal behavior is a highly worrying issue, with a difficult approach, which is even more complicated, if possible, in certain populations, which suffer circumstances that cause undoubted helplessness.
From the beginning, empirical work is approached from a qualitative perspective, of undoubted quality, which is reflected throughout the entire text.

I would like, however, to make suggestions for some clarification or change in the text that, I believe, would enrich what it provides. In this sense, I point out the following considerations:

-In reference to lines 50 - 58, some data should be added about the presence of significant differences regarding the differences between the figures noted. Although some of them have a substantial difference, this does not mean that these differences are significant.

-In reference to lines 62 - 63, a very blunt statement is made about the causes of suicide based on historical traumas derived from colonialism. Although evidence in this regard is drawn in previous paragraphs, I consider it necessary that this statement be well based on evidence, and supported by works that confirm it.

-Regarding lines 77-79, examples of the statements made should be provided.

-In relation to lines 92 - 93, why is the relationship that is stated clear?

Author Response

Thank you for the feedback regarding our manuscript Indigenous voices against suicide: A meta-synthesis advancing prevention strategies. Dr. Waters and I are very appreciative of the depth of engagement with our work and the feedback provided.   

We have responded to each of the comments, including a description of the content and location of changes in the revised manuscript. You will also see that we made some changes in the terminology used to better represent Indigenous populations in the United States, as ‘Indigenous’ or ‘Indigenous Peoples’, rather than Indigenous Peoples of the United States (IPoUS), per the guidance and recommendation of Indigenous scholars, mentors and peers.

In reference to lines 50 - 58, some data should be added about the presence of significant differences regarding the differences between the figures noted. Although some of them have a substantial difference, this does not mean that these differences are significant. 

Thank you for your note, however as biological sex was not specific to these analyses, national and state level prevalence data related to biological sex differences has been removed from the Introduction section.  

In reference to lines 62 - 63, a very blunt statement is made about the causes of suicide based on historical traumas derived from colonialism. Although evidence in this regard is drawn in previous paragraphs, I consider it necessary that this statement be well based on evidence, and supported by works that confirm it. 

We appreciate the reviewer for this note, and although we do cite literature that indicates the impact of historical trauma as a result of colonialism, we have edited this sentence to indicate a link, rather than a direct causation. Due to other reviewer revisions, you will see in lines 58-59 that the sentence now reads as follows: 

“Causal factors for suicide have been linked to historical trauma as a result of settler-colonialism, which severely influences the health of AI/AN and NH populations...”. 

Regarding lines 77-79, examples of the statements made should be provided. 

We appreciate this comment and an opportunity to clarify. You will find that we provided edits in lines 73-78, and the sentence now reads as follows:  

“Forced acculturation of Indigenous communities continues in the present with AI/AN and NH people consistently facing acts of discrimination and political silencing as well as an overall lack of belonging. AI/AN and NH people face consistent and ongoing acts of prejudice and discrimination, such as racist sports mascots, as well as lack of belonging, representation access, and support in social systems like schools, healthcare, and child welfare...” 

In relation to lines 92 - 93, why is the relationship that is stated clear? 

Many thanks to the reviewer for an opportunity to clarify. The term ‘clear’ was initially utilized to exemplify the literature as outlined in the paragraphs previously. However, we have revised the language used in this sentence for fluid readability. Due to other reviewer revisions, you will now see in lines 670-672 that the sentence now reads as follows:  

“The relationship between historical and perpetual trauma resulting from colonization must be acknowledged and investigated; experiences must be shared, and voices heard to shape culturally informed prevention strategies”.  

Round 2

Reviewer 1 Report

Comments and Suggestions for Authors

The authors have well revised based on the comments.